# A suite of selective pressures supports the maintenance of alleles of a *Drosophila* immune peptide

**Sarah R Mullinax[1], Andrea M Darby[2], Anjali Gupta[3], Patrick Chan[1], Brittny R Smith[1], Robert L Unckless[1]***

[1]Department of Molecular Biosciences, University of Kansas, Lawrence, United States; [2]Department of Entomology, Cornell University, Ithaca, United States; [3]Department of Ecology and Evolutionary Biology, University of Kansas, Lawrence, United States

## eLife Assessment

This **valuable** study investigates evolutionary aspects around a single amino acid polymorphism, known to be under long-term balancing selection, in an immune peptide of *Drosophila melanogaster*. Using alleles with different substitutions, the investigators demonstrate that while one allele provides better survival after systemic infections by a bacterial pathogen, the alternative allele endows its carriers with a longer lifespan under certain conditions. The authors suggest that these contrasting fitness effects of the two alleles contribute to balancing their long-term evolutionary fate. While the work is very interesting, the strength of the provided evidence is still **incomplete**, and the study would benefit from more rigorous approaches.

***For correspondence:**
unckless@ku.edu

**Competing interest:** The authors declare that no competing interests exist.

**Abstract** The innate immune system provides hosts with a crucial first line of defense against pathogens. While immune genes are often among the fastest evolving genes in the genome, in *Drosophila*, antimicrobial peptides (AMPs) are notable exceptions. Instead, AMPs may be under balancing selection, such that over evolutionary timescales, multiple alleles are maintained in populations. In this study, we focus on the *Drosophila* AMP Diptericin A, which has a segregating amino acid polymorphism associated with differential survival after infection with the Gram-negative bacteria *Providencia rettgeri*. Diptericin A also helps control opportunistic gut infections by common *Drosophila* gut microbes, especially those of *Lactobacillus plantarum*. In addition to genotypic effects on gut immunity, we also see strong sex-specific effects that are most prominent in flies without functional *diptericin A*. To further characterize differences in microbiomes between different *diptericin* genotypes, we used 16S metagenomics to look at the microbiome composition. We used both lab-reared and wild-caught flies for our sequencing and looked at overall composition as well as the differential abundance of individual bacterial families. Overall, we find flies that are homozygous for one allele of *diptericin A* are better equipped to survive a systemic infection from *P. rettgeri*, but in general have a shorter lifespans after being fed common gut commensals. Our results suggest a possible mechanism for the maintenance of genetic variation of *diptericin A* through the complex interactions of sex, systemic immunity, and the maintenance of the gut microbiome.

## Introduction

An effective immune response is essential for organisms to protect themselves from pathogens. However, an excessive immune response causes damage to self either directly (autoimmunity, cytokine storms) or through dysbiosis via disruption of the composition of beneficial microflora

(*Miyauchi et al., 2023*; *Wang et al., 2020*; *Badinloo et al., 2018*; *Fajgenbaum and June, 2020*). The dynamic suite of microbes that hosts face necessitates initiating a robust systemic immune response, avoiding self-harm, and maintaining a beneficial microbiome. The challenge of maintaining a balanced immune response is exacerbated because the threat of microorganisms is contextual; thus, the immune system must distinguish harmful vs. beneficial microbes in each context (*Thaiss et al., 2016*; *Shi et al., 2017*). For example, an ingested microbe may be harmless or even beneficial as a food source in the digestive tract, but, when in circulation, that same microbe may prove harmful to the organism (*Brown et al., 2012*). One consequence of balancing a robust innate immune system is that it may lead to the maintenance of genetic polymorphism in genes that encode for proteins intimately involved in the immune response (*Bagnicka et al., 2010*; *Pierce et al., 2021*). Such patterns of maintained polymorphisms contrast the standard coevolutionary arms race model where hosts and pathogens continually adapt to each other, leading to rapid evolutionary change in immune genes (*Anderson et al., 2010*; *Carrillo-Bustamante et al., 2015*; *Marques and Carthew, 2007*). Though the coevolutionary arms race model appears apt in many cases, the case for the adaptive maintenance of alleles, balancing selection in the broad sense, on genes involved in the immune defense stems from a more nuanced view of the delicate interplay of systemic immunity, life history traits, and the beneficial microbiome (*Liston et al., 2021*; *Koenig et al., 2019*).

Balancing selection is the general term for the adaptive maintenance of multiple alleles in a population. In contrast to the evolutionary arms race model, natural variation in many immune genes is maintained by balancing selection (*Chapman et al., 2019*; *Aguilar et al., 2004*; *Ferrer-Admetlla et al., 2008*; *Tennessen and Blouin, 2008*). The inference of balancing selection from molecular population genetic data is difficult, especially when genes are small and in areas of high recombination. Nonetheless, several examples of balancing selection exist, many of which involve small effector proteins of the innate immune system (*Tennessen and Blouin, 2008*; *Unckless et al., 2016*; *Chapman et al., 2016*; *Padhi and Verghese, 2008*).

Balancing selection on immune genes is likely to involve allelic benefits that are only conditionally beneficial. One way that an allele could be conditionally beneficial is if there is specificity between pathogen and allele (*Sommer, 2005*) such that *allele A* better protects against *pathogen 1* and *allele B* better protects against *pathogen 2: pathogen specificity hypothesis*. Another way that an allele can be conditionally beneficial in the context of immune defense is if resistance alleles are costly in the absence of infection. In this case, *allele A* better protects against both *pathogen 1* and *pathogen 2*, but in the absence of infection, *allele A* is costly for its host (*Williams et al., 2005*). This cost could be energetic or through autoimmune-like damage: *autoimmune hypothesis*. In reality, there is likely a continuum between these alternative hypotheses.

Invertebrates lack an adaptive immune system, so the delicate balance between systemic and gut immunity is achieved only through adjustments to the innate immune response (*Ryu et al., 2010*). It is therefore important to understand how invertebrates optimize their systemic immune response with as little detriment to their beneficial gut microbiota as possible.

Though first characterized for its role in systemic immunity, the (IMD) pathway is also the main NF-κB immune pathway in the gut and contributes to the maintenance of the microbiome and protection from gut infections (*Lhocine et al., 2008*; *Bosco-Drayon et al., 2012*; *Broderick et al., 2014*). To date, there is little research into how individual AMPs help maintain a healthy microbiome composition in the fly, let alone how allelic variation may affect microbiome composition (*Hanson and Lemaitre, 2020*; *Marra et al., 2021*). The standard gut microbiome of lab-reared *Drosophila melanogaster* consists of two prominent bacterial genera: *Lactobacillus* and *Acetobacter* (*Broderick et al., 2014*). Both genera are easily cultured in the lab, and the *Drosophila* gut can easily be manipulated, making it a good model to study the effects of immune gene variation on microbiome composition and how microbiome composition can, in turn, influence host fitness.

Antimicrobial peptides (AMPs) are a critical part of the innate immune system that act as broad-spectrum antimicrobials, combating bacteria, fungi, and viruses. In the coevolutionary arms race model, AMPs are on the front lines: directly interacting with microbes. AMPs have also been demonstrated to have critical roles in other physiological functions, e.g., dysregulation of AMPs has been connected to diseases such as atopic dermatitis and Alzheimer's disease (*Wang et al., 2020*; *Ong et al., 2002*; *Rieg et al., 2005*). AMPs also contribute to the aging process, where the immune system

becomes more active as organisms age to compensate for a decline in its effectiveness. However, this leads to cytotoxicity that in turn shortens lifespan (*Badinloo et al., 2018*).

In *Drosophila,* AMPs play a crucial role in both managing systemic infections and in the maintenance of the gut microbiota (*Buchon et al., 2014*). AMPs, however, appear to evolve more slowly than most immune gene families (*Clark et al., 2007*; *Jiggins and Kim, 2007*; *Sackton et al., 2007*; *Obbard et al., 2009*). One explanation for this perceived lack of adaptive evolution is that *Drosophila* AMPs' genetic variation is maintained adaptively through balancing selection (*Chapman et al., 2019*; *Comeron, 2014*; *Unckless and Lazzaro, 2016*; *Perlmutter et al., 2024*), where different selective pressures favor different alleles. Naturally occurring allelic variation in AMP loci is sometimes associated with variability in pathogen resistance (*Unckless et al., 2016*; *Lazzaro et al., 2004*; *Smith et al., 2023*; *Hanson et al., 2023a*). Diptericin A (hereon referred to as *diptericin* or *Dpt*) is one of the canonical effector genes of the *Drosophila* immune deficiency (IMD) pathway and is generally associated with defense against Gram-negative bacterial infection (*Tanji et al., 2007*; *Broderick, 2016*).

In both *D. melanogaster* and *Drosophila simulans*, an amino acid polymorphism at the 69th residue of the mature 83-residue Diptericin peptide segregates in most populations surveyed (*Unckless et al., 2016*). In both species, this polymorphism is due to a point mutation that changes the ancestral serine (S) allele to arginine (R), but the two species use different codons for arginine. *Unckless et al., 2016*, found a significant difference in survival based on *diptericin* genotype after systemic infection with the Gram-negative bacteria, *Providencia rettgeri*, a natural pathogen of *D. melanogaster*. The study used inbred fly lines and found homozygous serine flies have a better survival rate 5 days post infection (~60%) than homozygous arginine flies (~20%) or flies with a premature stop codon in *diptericin* (0%). Later work by Hanson et al. took a more general approach to AMP specificity and found that *Diptericin* plays a disproportionate role in response to infection with *P. rettgeri* (*Hanson et al., 2019aHanson et al., 2019a*). What remains unclear, however, is why this presumably deleterious arginine allele persists in populations of two different species. We hypothesize that it either protects against a different suite of pathogens or it is beneficial in the absence of infection through some life history trade-off.

This study aimed to test two hypotheses about the maintenance of genetic variation in *Diptericin*. First, different *Diptericin* alleles might protect against different pathogens – we refer to this as the *pathogen specificity hypothesis*. It would be supported if the arginine allele were to be associated with higher survival than the serine allele in some infections, since the serine allele is already clearly more protective against *P. rettgeri* (*Unckless et al., 2016*; *Hanson et al., 2023a*). In contrast, flies with the serine allele might be generally better at surviving infection, but this may have a cost to the organism. We refer to this as the *autoimmune hypothesis*, where a protective immune allele is associated with a trade-off that is costly in the absence of infection. We test whether there are trade-offs between systemic immunity and other life history traits – particularly traits related to the gut microbiome utilizing genetically controlled lines for *Diptericin* alleles generated by CRISPR/Cas9 editing. Flies were systemically infected with a panel of six different bacteria, and homozygous serine flies had a better 5-day survival with each bacterium. We then used axenic and gnotobiotic flies to look at the lifespan of flies since AMP overexpression and microbial proliferation are commonly observed in aging flies (*Arias-Rojas and Iatsenko, 2022*; *Badinloo et al., 2018*; *Shit et al., 2022*; *Hanson and Lemaitre, 2023b*). *Lactobacillus plantarum* is harmful to female flies with nonfunctional diptericin, while homozygous arginine flies poly-associated with *L. plantarum* and *Arctocephalus tropicalis* had a longer lifespan than homozygous serine flies. In this way, we found evidence for a trade-off between the ability to fight systemic infection and the ability to control opportunistic gut infections.

## Results
### A single amino acid change drastically influences survival after infection

*Unckless et al., 2016*, found that in inbred lines from the *Drosophila* Genetic Reference Panel (DGRP), flies homozygous for serine as position 69 of the mature Diptericin peptide survive systemic infection with *P. rettgeri* much better than those homozygous for the arginine peptide at the same position. This experiment showed only an association between immune defense and the serine/arginine polymorphism since these inbred lines were on several different genetic backgrounds. To control genetic background, we used CRISPR/Cas9 genome editing to create both an arginine allele (single

nucleotide change, $dpt^{S69R}$) and multiple null alleles (1 or 3 base pair deletions, $\Delta dpt$ flies), as well as control $dpt^{S69}$ (serine at position 69 of the mature peptide) in *diptericin* (*Figure 1—figure supplement 1A*, *Supplementary file 1*). The phenotype for our CRISPR/Cas9-edited flies showed striking similarity to the inbred lines. In systemic infection challenges with *P. rettgeri*, $dpt^{S69}$ flies are better protected from infection than $dpt^{S69R}$ flies (p=5.42e-08) and $\Delta dpt$ flies (p=6.46e-09, *Figure 1A*). Remarkably, for inbred lines and CRISPR/Cas9-edited lines, survival for 5 days post infection with *P. rettgeri* for the serine allele lines is 50–60%, with the arginine allele is 10–20%, and less than 5% for null alleles.

We next challenged the CRISPR/Cas9-edited flies with systemic infection using multiple other Gram-positive and Gram-negative bacteria to determine whether the arginine allele ($dpt^{S69R}$) protects against some infections better than the serine allele. Such a finding would support the hypothesis that allelic variation is maintained by different alleles providing specific protection against different microbes. However, the differential response to systemic infection with *P. rettgeri* at an $OD_{600}=0.1$, as described above, remains the largest difference in immune response between *Dpt* genotypes. $Dpt^{S69}$ flies survived better than $dpt^{S69R}$ flies for all systemic infections tested (*Figure 1*, *Supplementary file 1*). In the Gram-positive bacterial infections (*E. faecalis*, *S. succinus*, *L. fusiformis*, and *L. lactis*), $dpt^{S69R}$ flies had lower survival than that of $\Delta dpt$ or $imd^-$ (null allele for the Imd gene) flies (*Figure 1B–E*). The only systemic infection where *dpt* genotype did not seem to matter was *S. marcescens*, where all flies survive infection well except for $imd^-$, which die very quickly (*Figure 1F*).

We also tested if males and females showed differences in systemic immunity based on genotype for *P. rettgeri*, *E. faecalis*, and *L. plantarum* infections (*Figure 1—figure supplement 2*, *Supplementary file 3*). Generally, the *Dpt* genotypes have qualitatively similar survival patterns between males and females when infected with *P. rettgeri*. Females showed lower survival compared to males after systemic infection with *E. faecalis* (p=0.0003). Higher male survival post infection was observed previously for both *E. faecalis* and *P. rettgeri*, with differences ascribed to the Toll pathway (*Duneau et al., 2017*).

Overall, flies with the $dpt^{S69}$ allele are better equipped to survive a systemic bacterial infection than $dpt^{S69R}$ flies, though this is most pronounced for *P. rettgeri*. Of the six bacteria tested, there was no case where $dpt^{S69R}$ flies survived the infection better than $dpt^{S69}$ flies. These results do not support the hypothesis that alleles are maintained to better combat different pathogens.

## Diptericin genotype affects the lifespan of mono- and poly-associated gnotobiotically reared flies

Although our survey of systemic infections was not exhaustive, we did not find any instances where $dpt^{S69R}$ flies were better able to fight infection, so we turned our attention to the role of *Dpt* in gut microbiome maintenance and immunity. The gut microbiome influences several life history traits in *Drosophila* and other organisms (*Ding et al., 2019*; *Ludington and Ja, 2020*; *Erkosar and Leulier, 2014*; *Macke et al., 2017*). To dissect how *diptericin* genotype influences microbiome maintenance, we manipulated the microbiota in CRISPR/Cas9 flies and measured longevity and bacterial load. We began with the longevity of axenically reared flies, since it represents a baseline survival without the presence of microbes. Flies with functional copies of *diptericin* ($dpt^{S69}$ or $dpt^{S69R}$) did not differ in lifespan for either sex ($p_{genotype}=0.3762$), and $\Delta dpt$ lines show a similar lifespan. However, there was a difference in overall lifespan between the sexes among the CRISPR genome-edited lines ($p_{sex}=1.04e-6$). In $dpt^{S69}$ or $\Delta dpt$, female flies have a longer lifespan than male flies, as observed generally for *D. melanogaster* previously (*Lints et al., 2009*). In contrast, $imd^-$ male flies have a much longer lifespan than any of the other lines tested (*Figure 2A* – axenic row, *Supplementary file 3*).

We next tested the influence of Diptericin genotype on mono-associations with standard constituents of the *Drosophila* gut microbiome. We found multiple sex effects when axenically reared flies were fed *L. plantarum*, *L. brevis*, or *A. tropicalis* (*Figure 2A and B*, *Figure 2—figure supplement 1*). Flies that were fed *L. plantarum* show the most striking differences. Male flies continued to show similar lifespans to each other, but female $\Delta dpt$ and female $imd^-$ flies both succumbed quickly post-feeding, indicating functional *Diptericin* is important for gut immunity against opportunistic *L. plantarum* in females.

We also fed axenic flies *P. rettgeri*, which does not readily cause gut infection, to axenically reared flies. Recall that after systemic infection, $dpt^{S69}$ flies are more resistant to *P. rettgeri* than $dpt^{S69R}$ flies (*Figure 1A*). After mono-association, however, we see no significant difference between the two

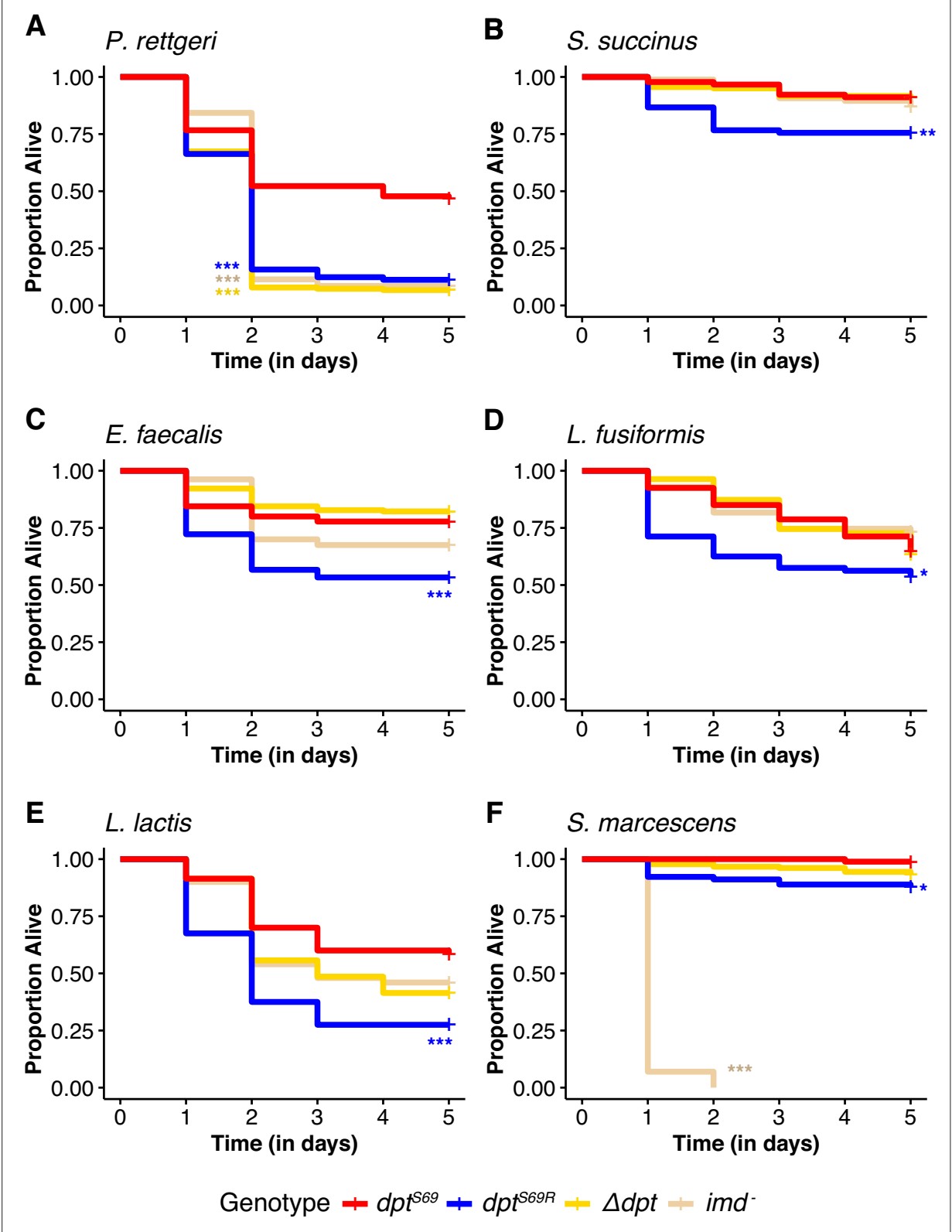

**Figure 1.** Five-day survival post systemic infection. Survival after systemic immunity challenge with the following bacteria: (**A**) *P. rettgeri,* (**B**) *Staphylococcus succinus,* (**C**) *Enterococcus faecalis,* (**D**) *Lysinibacillus fusiformis* Strain Juneja, (**E**) *Lactococcus lactis,* and (**F**) *Serratia marcescens.* Each graph represents the combined results of three different infection dates of at least 20 males of each genotype for each date (at least 60 total).

*Figure 1 continued on next page*

*Figure 1 continued*

Significance is relative to dpt$^{S69}$ (red line) using Cox proportional hazards regression model. *p<0.05, **p<0.01, ***p<0.001. Each sample n>60 per genotype.

The online version of this article includes the following figure supplement(s) for figure 1:

**Figure supplement 1.** CRISPR/Cas9 genome editing of the *dpt* locus.

**Figure supplement 2.** Male vs. female survival after systemic infection.

genotypes with *P. rettgeri* in either sex (p=0.612, *Figure 2B*). There is also no significant difference between the *Δdpt* and lines with functional Diptericin (*Figure 2B*, *Figure 2—figure supplement 1*). Surprisingly, however, the null allele for *Imd* again shows a significant effect on survival in a sex-specific manner: null *Imd* females die much earlier, and null *Imd* males survive much longer when fed *P. rettgeri*.

Some of the sex-specific differences in survival after mono-association with bacteria may be driven by intrinsic differences in feeding rates between the sexes (and potentially genotypes as well) (*Wu et al., 2020*). To determine whether the differences in male and female survival and load were due in part to differences in feeding rates, we performed a feeding rate assay with blue dye mixed with media (Luria-Bertani broth) or *P. rettgeri* (OD$_{600}$=15.0). We noted that females did eat more in a single hour of feeding, but that there was also an effect of *diptericin* genotype (with null flies eating less, *Figure 2—figure supplement 2*). Thus, it is possible that differences in longevity after mono-association are due to different rates of exposure to those bacteria because of different feeding rates.

We next looked at how poly-associations with bacteria affected lifespan. First, we fed flies a 1:1 mixture of *L. plantarum* and *A. tropicalis*, two common gut microbes found in lab-reared and wild-caught flies (*Wong et al., 2011*; *Bost et al., 2018*). We found that female *Δdpt* and *imd*⁻ flies had a much shorter lifespan than flies with functional Diptericin (*Figure 2A and B*). This is the same pattern observed in mono-association with *L. plantarum*. However, we observe that the *diptericin* genotype influences survival when poly-associated with *L. plantarum* and *A. tropicalis*. Dpt$^{S69R}$ female flies live longer than *dpt*$^{S69}$ female flies (*Figure 2A and B*; p=0.00782). This is the opposite of systemic infections, where *dpt*$^{S69}$ flies always survived better than *dpt*$^{S69R}$ flies, and may indicate a trade-off between defense against systemic and gut immunity.

In poly-association with 1:1 *L. plantarum* and *P. rettgeri,* we observe many of the same patterns as in the poly-association with *L. plantarum* and *A. tropicalis*. Again, female *Δdpt* and *imd*⁻ flies have a shorter lifespan than female *dpt*$^{S69}$ and *dpt*$^{S69R}$ flies (*Figure 2B*), and *dpt* genotype is associated with lifespan. Dpt$^{S69R}$ females have a longer lifespan than *dpt*$^{S69}$ flies (p=0.000198, *Figure 2B*).

Given the genotypic and sex effects on survival after oral association with *L.* plantarum, we also looked at sex differences after systemically infecting conventionally reared flies with *L. plantarum*. We saw no difference in survival between the sexes, as observed for systemic infections with *E. faecalis* or *P. rettgeri,* and saw no difference between *Δdpt* lines and lines with functional Diptericin, as observed for axenic flies mono-associated with *L. plantarum* (*Figure 1—figure supplement 2*). This could indicate that Diptericin plays different roles for systemic and gut immunity in relation to *L. plantarum* in each sex. Further, *dpt*$^{S69}$ male flies survived better than *dpt*$^{S69R}$ male flies (p=0.00438), in line with observations from other systemic infections in males (*Figure 1*).

Overall, we found a role of both specific *diptericin* genotype (serine vs. arginine) and the presence of functional copies of *diptericin* for survival after introducing common gut microbes in controlled conditions. Most striking was the sexually dimorphic role of both *Dpt* and *Imd*⁻, with females being much more sensitive to genotype than males.

## Gnotobiotic fly bacterial load

Given the differences in survival among sexes and genotype for different gnotobiotic associations with bacteria, we determined whether bacterial load after associations also was different among sexes and genotypes. To assess the influence of genotype on the immune response of aging flies, we studied how well common gut bacteria colonized the gut over 20 days, representing ~20–33% of *D. melanogaster's* normal lab lifespan (*Giannakou and Partridge, 2007*; *Biteau et al., 2011*). We generated gnotobiotic flies by feeding specific bacteria for 2 days and observed the bacterial load 2, 10, and 20 days post feeding.

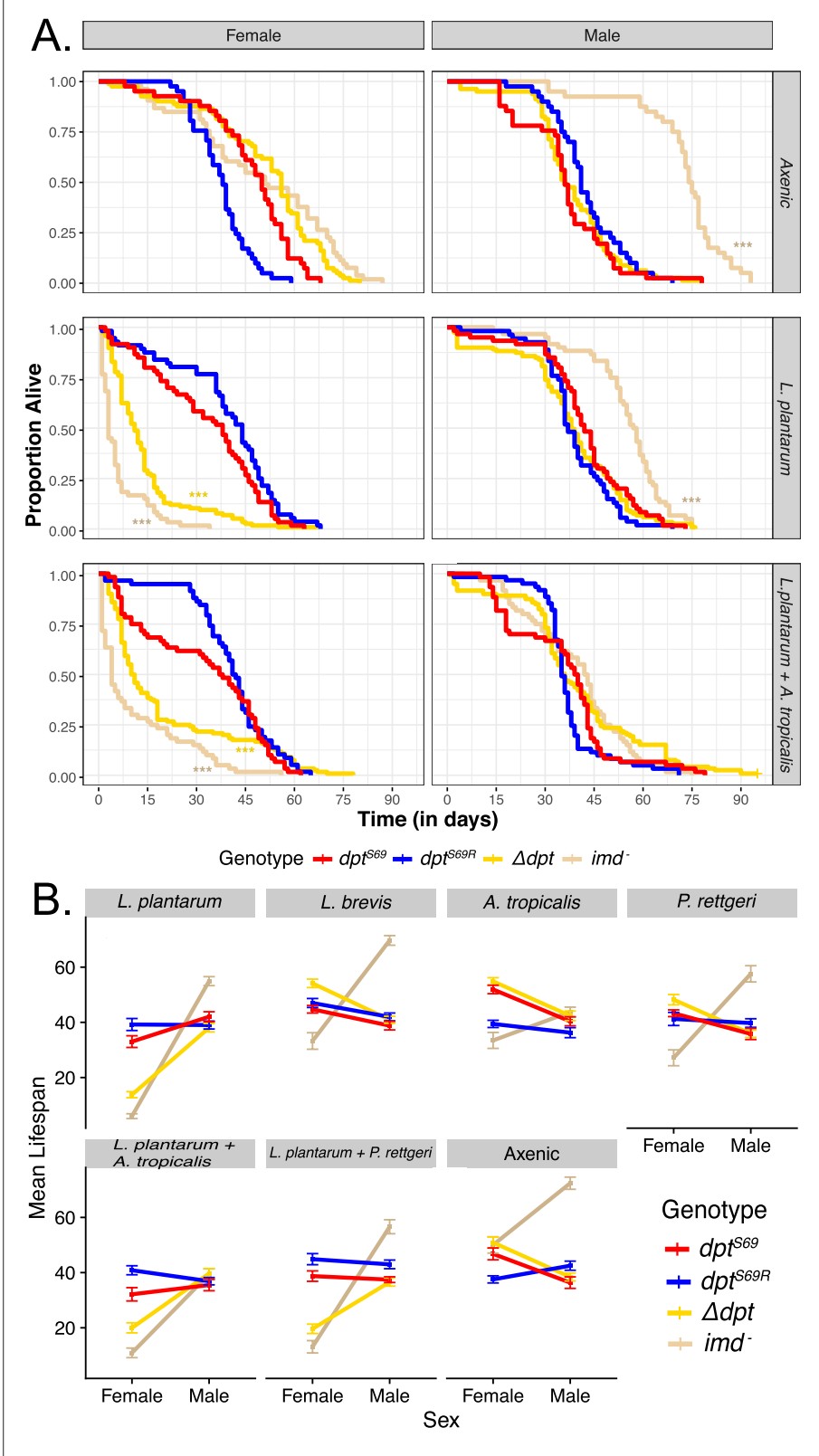

**Figure 2.** Axenic and gnotobiotic lifespan. (**A**) Survival plots for axenic flies and gnotobiotic flies mono-associated with *L. plantarum* or poly-associated with *L. plantarum* and *A. tropicalis*. Each curve represents 60 flies from 3 replicates of 20 flies. Significance is shown in comparison to SS (red line) using the model Lifespan~(Genotype*Sex)/Vial + Block. (**B**) Mean lifespan was plotted against each sex and separated by bacterial

*Figure 2 continued on next page*

*Figure 2 continued*

association. Axenic flies have no bacterial association. Each point for each sex in a combination of 3 separate trials of 20 flies for a total of 60 flies. These interaction plots only have mean lifespan and thus are useful for seeing all the data at a glance. Error bars are mean plus or minus the standard error of the mean. *p<0.05, **p<0.01, ***p<0.001.

The online version of this article includes the following figure supplement(s) for figure 2:

**Figure supplement 1.** Gnotobiotic longevity.

**Figure supplement 2.** Feeding rates.

When specifically looking at mono-association with *L. plantarum,* we observe that bacterial load differences between genotypes occur within the first 2 days post feeding but begin to disappear by day 20 ($p_{genotype(day2)}$=2.6e-5, $p_{genotype(day20)}$=0.709, *Figure 3A*, *Figure 3—figure supplement 1*). Note that in the first 2 days, the flies were raised on microbe-contaminated media, but after 2 days were moved onto sterile food and then transferred to new sterile food every 3 days. This corroborates what we saw in the longevity data in females. Within the first 15 days, a large proportion of $\Delta dpt$ female flies died, and we observed a higher bacterial load in these flies, especially on day 2 post feeding (*Figure 2*). By day 20, differences in bacterial load in females disappeared. Note that there is inherent sampling bias, as only the flies able to survive until day 20 are sampled at that time point. In the case of $imd^-$ flies, no females survived until day 20; hence, there is no data for $imd^-$ flies on day 20 (*Figure 3A*, *Figure 3—figure supplement 1*).

*P. rettgeri* is the only canonical systemic pathogen we included in our gnotobiotic study, and the patterns based on genotype were different between systemic infection and the gnotobiotic oral infection data. When flies were fed *P. rettgeri,* we saw a range of bacterial loads, from zero colonies to bacterial load levels on par with common gut bacteria. We also observed a genotype effect between $dpt^{S69}$ and $dpt^{S69R}$ on day 10 and 20 post feeding in males ($p_{genotype(day10)}$=0.0044, $p_{genotype(day20)}$=0.0004, model only included $dpt^{S69}$ and $dpt^{S69R}$, *Supplementary file 5*). In both instances, bacterial load in $dpt^{S69R}$ flies is higher than in $dpt^{S69}$ flies. This may indicate that $dpt^{S69}$ flies are better equipped to deal with both systemic and oral infection from *P. rettgeri*. Whether this is due to a greater ability of $dpt^{S69}$ flies to withstand the effects of infection (tolerance) by *P. rettgeri* remains a question. It is also important to note that *P. rettgeri* establishes poorly in the gut of wildtype flies (personal observation), which may explain the noisy results for mono-associations after oral infection.

A range of bacterial loads was also observed when flies were poly-associated with *L. plantarum* and *P. rettgeri* (*Figure 3B*, *Figure 3—figure supplement 2*). There were no statistically significant differences between $dpt^{S69}$ and $dpt^{S69R}$ bacterial for this poly-association (*Supplementary file 6*). In fact, there were no significant differences between any of the *P. rettgeri* bacterial loads; however, *L. plantarum* does show differences on day 2 post feeding (p=7.97e-6). These differences are mainly between flies with nonfunctional *diptericin* ($imd^-$ and $\Delta dpt$) and flies with functional *diptericin*. Both $\Delta dpt$ and $imd^-$ flies had higher bacterial load than $dpt^{S69}$ and $dpt^{S69R}$ flies on day 2 post feeding, which may be an indication of the reason both these lines quickly succumb to feeding with *L. plantarum*, at least in the context of the poly-association with *P. rettgeri*.

We observed larger differences in bacterial load shortly after feeding with bacteria, and those differences became less by day 20 post feeding. We did not see any differences in *P. rettgeri* load when mono-associated or when part of a poly-association, indicating the flies respond differently to the same pathogen when introduced systemically or orally.

## Evidence for life history trade-offs mediated by *diptericin* genotype

Proteins involved in a robust immune response may have pleiotropic effects on other traits either because of a direct interaction with that trait or because of inherent costs of immune defense (self-damage, energy expenditure, etc.) (*Viney et al., 2005*; *Sadd and Siva-Jothy, 2006*; *Lin et al., 2018*). We examined three life history traits (desiccation stress survival, starvation stress survival, and uninfected longevity) to determine whether *Dpt* genotype had such pleiotropic effects.

The ability to survive desiccation stress is an important life history trait for wild *Drosophila* survival (*Gibbs, 2002*; *Wang, 2017*). When subjecting our male CRISPR/Cas9 flies to desiccation, we observe conventionally reared flies succumb faster to desiccation stress than axenically reared flies (*Figure 4A*,

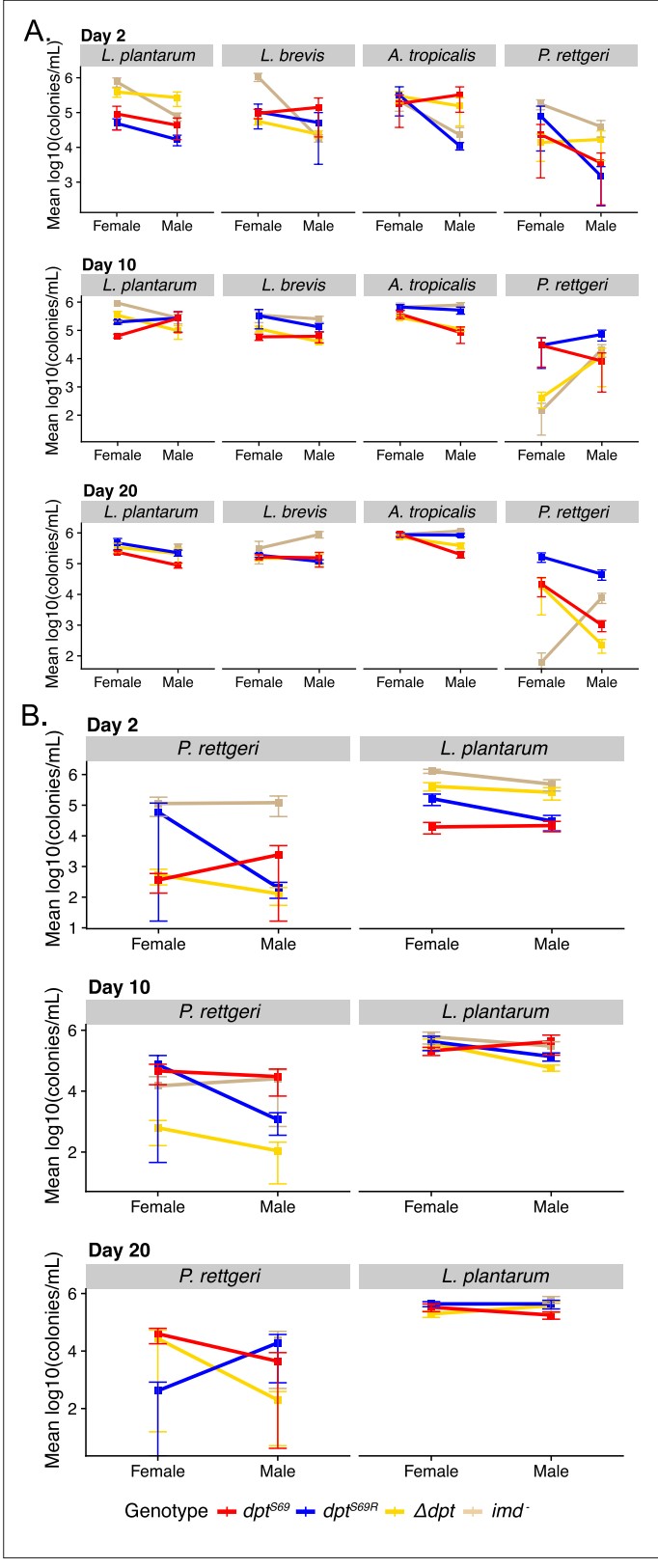

**Figure 3.** Bacterial load interaction plots for day 2, 10, and 20 post association with various microbes. Two-day-old flies were placed on food seeded with a bacterial suspension of each bacteria at an $OD_{600}$ of 15. Each point for each sex is from 3 separate trials of 2 samples of 3 flies each for a total of 6 samples per point. (**A**) Mono-associations where each plot represents a different condition, (**B**) poly-associations where each row plots both

*Figure 3 continued*

the *P. rettgeri* and *L. plantarum* load from the same flies but were plated on LB and De Man, Rogosa, and Sharpe (MRS), respectively. Error bars are mean plus or minus the standard error of the mean.

The online version of this article includes the following figure supplement(s) for figure 3:

**Figure supplement 1.** Bacterial load at day 2, 10, and 20 days post feeding.

**Figure supplement 2.** Bacterial load of *L. plantarum* and *P. rettgeri* poly-association.

$p_{genotype} < 2e{-}16$). In conventionally reared males, $dpt^{S69}$ flies have similar desiccation resistance to $dpt^{S69R}$ flies (p=0.917). However, $dpt^{S69}$ flies survive desiccation stress better than $dpt^{S69R}$ flies when reared axenically ($p_{genotype}=0.04126$). We also compared *Drosophila* OreR and W1118 (both homozygous for the serine allele of Diptericin) to the $dpt^{S69}$ line (*Figure 4—figure supplement 1*, *Supplementary file 7*). Unsurprisingly, despite the same *diptericin* genotype, all 3 lines show dramatically different desiccation resistance. Therefore, *diptericin* genotype plays limited, if any, role in variation in desiccation resistance, as all 3 wildtype lines were derived from different genetic backgrounds.

We next looked at the effect of genetic variation in *diptericin* on male ability to survive under starvation stress. As with desiccation resistance, we found axenically reared flies survive starvation stress better than conventionally reared flies (*Figure 4B*, *Figure 4—figure supplement 2*, $p_{treatment}<2e{-}16$). However, unlike the desiccation stress conditions, both conventionally reared and axenically reared $dpt^{S69R}$ flies survive starvation stress longer than $dpt^{S69}$ flies (conventional: $p_{genotype}<2e{-}16$, axenic: $p_{genotype}<2e{-}16$). In conventionally reared flies, there is no difference in survival between $dpt^{S69}$ flies and $\Delta dpt$ flies (*Figure 4B*, *Supplementary file 8*). However, in axenically reared flies, $\Delta dpt$ flies have an intermediate survival phenotype between $dpt^{S69}$ and $dpt^{S69R}$ flies. This may suggest an interaction between functional *diptericin* and the microbiome that influences starvation.

Finally, we looked at the overall longevity of female and male conventionally reared flies (in the absence of any infection or other significant selection pressure). Female flies have a longer lifespan than males (*Figure 4C*, *Figure 4—figure supplement 3*, $p_{sex}=5.80e{-}13$) regardless of *dpt* genotype. In males, $dpt^{S69}$ flies had a significantly longer lifespan than male $dpt^{S69R}$ flies (mean of 60.3 days for $dpt^{S69}$ and 54.1 days for $dpt^{S69R}$, p=0.0072), but not in female flies (mean of 61.9 days for $dpt^{S69}$ and 59.0 days for $dpt^{S69R}$, p=0.6434). However, in axenically reared flies, only female $dpt^{S69}$ flies have a longer lifespan than male $dpt^{S69R}$ flies. Interestingly, the female flies with the longest lifespan have nonfunctional Diptericin (*Figure 4C*, $\Delta dpt$ line and $imd^-$ line, *Supplementary file 9*). Males lacking functional Diptericin show the same effect, but to a lesser extent. These results provide evidence that, in the presence of a standard gut microbiota, both *Dpt* and a functional Imd pathway may decrease longevity. This is consistent with others who found downregulation of NF-κB pathways and AMPs increased lifespan in *Drosophila* (*Lin et al., 2018*; *Khor and Cai, 2020*), but the fact that flies with *Dpt* null alleles alone are sufficient to increase lifespan is noteworthy.

## Diptericin's influence on gut microbial diversity

To determine whether *Diptericin* genotype influences the composition of the bacterial community in the gut, we sequenced amplicons of 16S ribosomal rRNA in conventionally reared lab flies under two different rearing conditions: flies reared in standard *Drosophila* vials, and the progeny of the cross between $dpt^{S69}$ and $dpt^{S69R}$ flies reared in cages for more than two generations. The vials may capture the long-term impact of genotype on microbiota but make ruling out stochastic changes in communities difficult. The cages ensure identical microbiota during development, but do not allow gradual effects of genotype on microbiota to accumulate. First, we found flies that were co-reared in cages had similar microbiomes, with no discernible differences by the alpha diversity metric, Shannon diversity (p=0.5239, *Figure 5A*). On the other hand, the microbiomes of flies reared in vials were distinctly different overall compared to the microbiomes of flies reared in the cages (p<0.001, *Figure 5B*). However, the differences between genotypes were still minimal in the vial-reared flies, and vial may be a large factor in differences between lines. There were 2 $dpt^{S69R}$ lines used, and even though these are the same genotype and genetic background, there was a significant difference between the 2 lines (*Figure 5B*). This difference is likely due to the within-vial drift in gut microbiome communities (*Wong et al., 2013*), which we attempted to control for using flies of different genotypes reared in the same cage (*Figure 5A*).

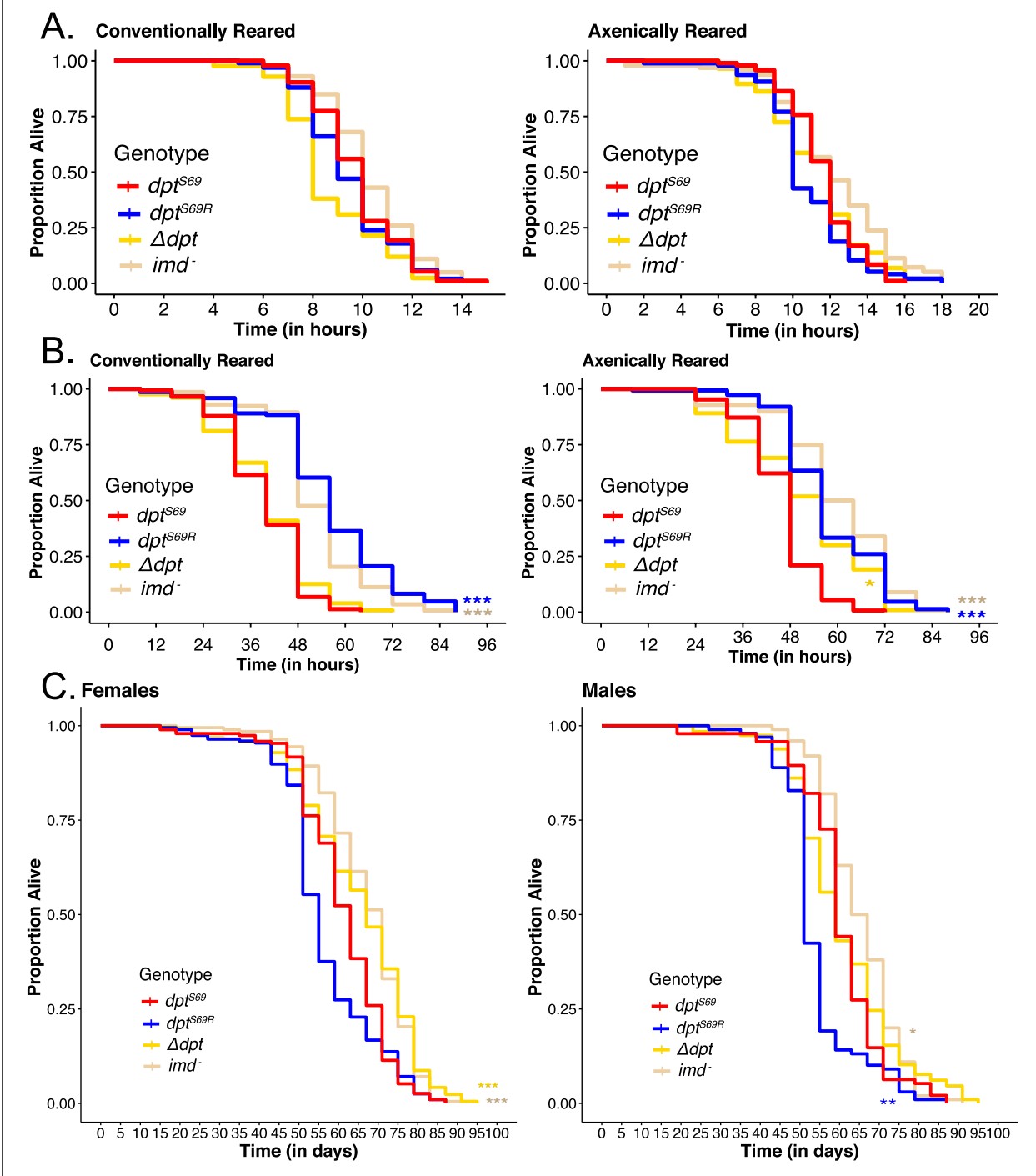

**Figure 4.** *Dpt* genotype is associated with variation in some life history traits. (**A**) Conventionally and axenically reared male flies desiccation resistance (N>90 for each line apart from *Δdpt* line). There is no significance between *dpt^S69* (red line) and other genotypes. (**B**) Conventionally and axenically reared male flies starvation resistance (N>90 for each line apart from *Δdpt* line). (**C**) Female and male lifespan for conventionally reared flies. N>93 for each line and sex. Significance is shown in relation to *dpt^S69* (red line). Note that axenic longevity is shown in ***Figure 2***. *p<0.05, **p<0.01, ***p<0.001.

The online version of this article includes the following figure supplement(s) for figure 4:

**Figure supplement 1.** Desiccation stress survival with additional wildtype lines (OreR and W1118).

**Figure supplement 2.** Starvation with additional wildtype lines (OreR and w1118).

**Figure supplement 3.** Longevity with separated *Δdpt* lines and heterozygotes.

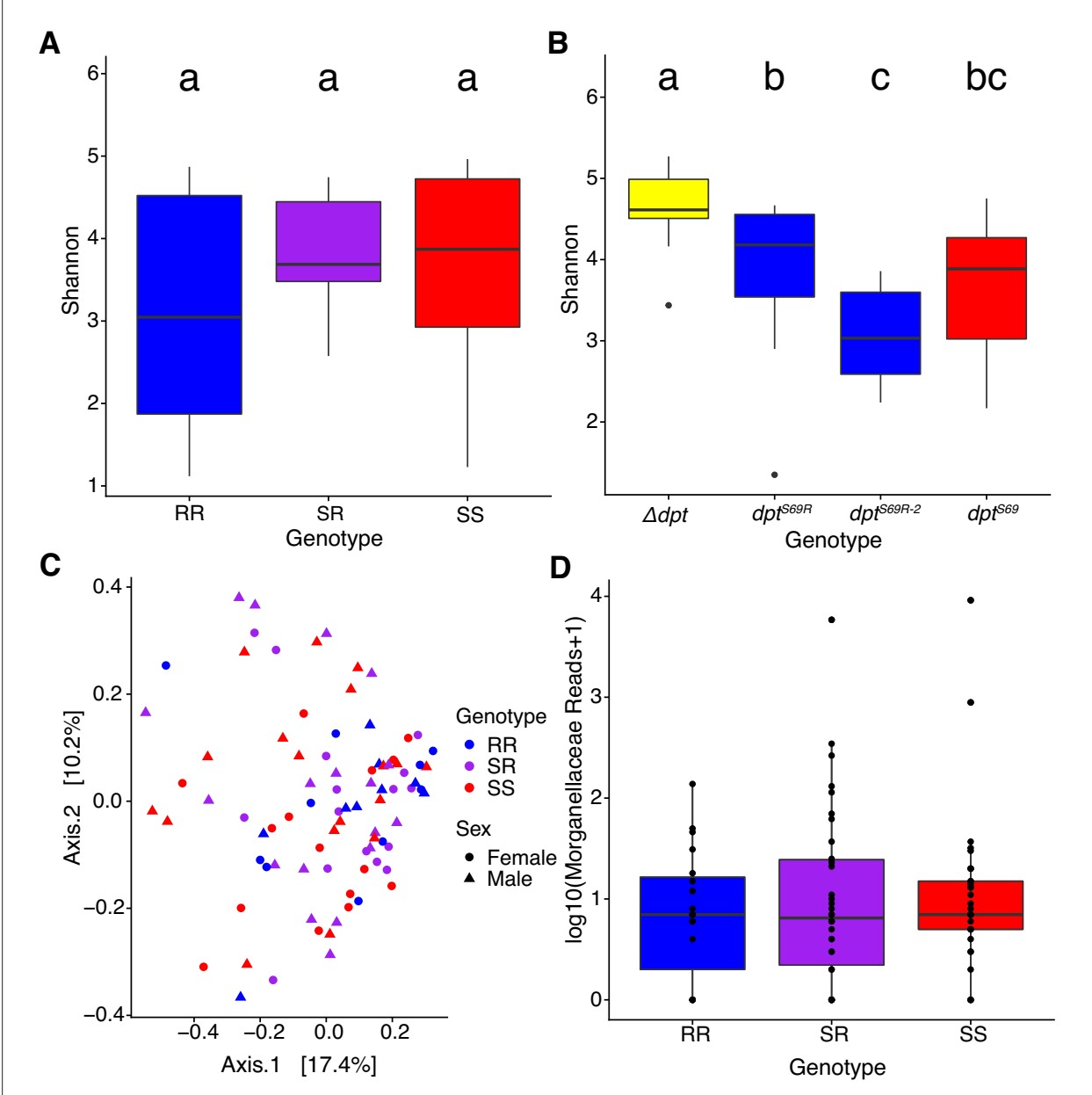

**Figure 5.** The influence of *Dpt* genotype on the gut microbiota. (**A**) Shannon diversity of flies co-reared in cages. (**B**) Shannon diversity of individual genotypes reared in vials. (**C**) Bray-Curtis dissimilarity of wild-caught flies. (**D**) Differential abundance of Morganellaceae in wild-caught flies. Different letters show significance based on post hoc Tukey's test, $p < 0.05$. Boxes represent 25th and 75th percentiles with the median denoted as a horizontal line in the middle of the box. Whiskers represent minimum and maximum of the data.

The lab is a controlled environment that allows for excellent control of variables but does not replicate the conditions found in nature. Thus, we looked at the microbiomes of wild-caught flies collected from the decaying fruit of an apple orchard. We found flies with homozygous for *dpt*[S69R], homozygous for *dpt*[S69], and heterozygous genotypes at our collection location (***Supplementary file 10***). Out of the 955 *D. melanogaster* flies successfully genotyped, only 20 flies homozygous for *dpt*[S69R] were identified. All 20 homozygous *dpt*[S69R] flies were profiled along with 36 each of *dpt*[S69] homozygous flies and heterozygous flies for a total of 92 flies profiled (amplicon sequences of both *Dpt* and 16S rRNA are in ***Supplementary file 2***).

We found that *dpt* genotype does not affect the overall composition of the microbiome of wild-caught flies based on Shannon diversity (alpha diversity) and Bray-Curtis dissimilarity (beta diversity)

(*Figure 5C*). However, when looking at the differential abundance of individual bacterial families, there are differences based on genotype. We tested the specific association between *diptericin* genotype and normalized counts from nine microbial families. Most intriguing is the difference in abundance of Morganellaceae family reads since *P. rettgeri* belongs to this family. Heterozygous and homozygous serine flies have a slightly higher abundance than homozygous arginine flies (*Figure 5D*), but these differences were not significant.

Overall, our results for associations between *Dpt* genotype and gut microbiome in both lab and wild flies are weak. Such associations may require much larger sample sizes if the effects are small – particularly given the noisy phenotype in the wild. Alternatively, the influence of *Dpt* genotype on gut microbiome diversity may be specific to developmental life stages or microbes that are relatively rare in these populations.

## Discussion

The adaptive maintenance of multiple alleles of a single gene has been posited for nearly a century (*Fisher, 1930*; *Ford et al., 1940*; *Dobzhansky, 1950*; *Cain and Sheppard, 1954*; *Dobzhansky, 1970*), and evidence for its pervasiveness continues to grow (*Hedrick and Thomson, 1983*; *Subramaniam and Rausher, 2000*; *Koskella and Lively, 2009*; *Leffler et al., 2013*; *Delph and Kelly, 2014*; *Lee et al., 2021*). However, there are relatively few cases where we understand the adaptive benefits of both alleles (sickle cell anemia and malaria resistance being the best studied example [*Piel et al., 2010*]). In this study, we generated CRISPR/Cas9 genome-edited flies of different *diptericin* genotypes to investigate mechanisms of maintenance of allelic variation in AMPs. This allowed us to specifically study a single amino acid change on an otherwise genetically controlled background. Overall, we found evidence that $dpt^{S69}$ flies survive systemic infection better, while $dpt^{S69R}$ flies survive some opportunistic gut infections better. Thus, more robust defense against systemic infection appears to interfere with maintaining a balanced gut microbiota – this trade-off may result in natural populations maintaining both alleles. Importantly, we note that while our results are consistent with the adaptive maintenance of alleles in natural populations, these lab assays do not prove the adaptive value of different alleles in the field.

Systemic infections with *P. rettgeri* in CRISPR/Cas9 genome-edited lines confirmed the amino acid polymorphism in Diptericin as the basis for the difference in immune response of flies with different *diptericin* genotypes previously hypothesized via an association study (*Unckless et al., 2016*). Surprisingly, $dpt^{S69}$ flies had a higher survival 5 days post infection than $dpt^{S69R}$ flies in *all* systemic infections tested. This may mean either that the serine allele provides improved defense against all the tested systemic infections, or that alterations to other aspects of fly physiology (notably the gut microbiota) could predispose arginine flies to poorer survival. It is also possible that some of *diptericin's* influence on survival is mediated by changes in the microbiome. We did not perform systemic infections on axenically reared flies, but if *Dpt* genotype influences the microbiome, which in turn influences survival after infection with other pathogens, *Dpt* should not influence survival after systemic infection if flies are reared axenically.

Our study also reveals an important role of Diptericin in preventing opportunistic gut infections by common gut microbes. This is especially evident when looking at lifespan after associating axenic flies with *L. plantarum* in both mono- and poly-association contexts. In both sexes, $dpt^{S69R}$ flies have a longer lifespan than $dpt^{S69}$ flies, and this effect is enhanced in poly-association with *L. plantarum* and *A. tropicalis*. Other studies have posited a role for individual AMPs in the maintenance of gut microbes in *Drosophila* (*Marra et al., 2021*), but this appears to be the first example linking a functional copy of a single AMP to survival differences.

One of our most striking and unexpected results was the distinct sex differences in longevity after association with common gut bacteria. Furthermore, fly genotype plays a significant role in these differences. After association with *L. plantarum*, $imd^-$ females had the shortest mean survival, while $imd^-$ males had the longest mean survival. The effect for $\Delta dpt$ was similar: females had much shorter average survival than the other two *dpt* alleles, but the three alleles showed equivalent longevity in males. While the $imd^-$ sexual dimorphism is relatively consistent across experiments with different microbes, the $\Delta dpt$ effect seems to be more limited to associations involving *L. plantarum*. This result not only highlights the importance of using both sexes in microbiome research, but also confirms that functional Diptericin is important for *D. melanogaster* gut health. There is a growing body of literature

on the effects of sexual dimorphisms in immune response, but it is lacking for sexual dimorphisms in gut immunity, and our findings only emphasize the need to fill this gap (*Drosophila* immunity sexual dimorphisms reviewed in *Belmonte et al., 2020*; also see *Shianiou et al., 2023*). Our feeding assay provides some evidence that the differences between male and female survival after exposure to microbes are due to the amount of food (and therefore the number of microbes) consumed, but these differences should be examined in much more detail. More generally, the sex by genotype effect suggests a likely explanation for the maintenance of genetic variation: different fitness effects of alleles in the different sexes (*Connallon and Clark, 2014*). Note that while our study focuses on the serine/arginine polymorphism, at least six null alleles of *dpt* segregate in natural populations of *D. melanogaster,* and the frequency of those nulls shows clinal variation in both North America and Africa (*Hanson et al., 2019b*); thus, the nulls are likely maintained selectively too.

Finally, we note that we have framed our argument in terms of balancing selection in the broad sense because we are agnostic as to the population dynamics upon which these different selective pressures might act. For example, it may be that some populations are disproportionately infected by *P. rettgeri* or other microbes for which the serine allele is favored, while other populations are relatively pathogen-free and the arginine allele is favored. This scenario is partly supported by the evidence of clinal patterns in allele frequency in Africa and (to a lesser extent) North America (*Hanson et al., 2019a*). Similarly, these selective pressures could vary temporally. On the other hand, these pressures could act all in one population where serine is favored in infected individuals, and the arginine allele is favored in uninfected individuals.

Herein, we highlight three main roles of Diptericin in *Drosophila*: (1) Diptericin genotype influences systemic immune defense; (2) in certain conditions, Diptericin genotype influences gut immunity; and (3) Diptericin has sex-specific effects in the gut. These results highlight the need for individuals and populations to modulate the immune system to balance systemic and gut immunity, and how the different needs of females and males complicate this balance. The dramatic differences in survival of males and females in response to oral infection by common gut bacteria underline the importance of looking at both sexes when examining the maintenance of genetic diversity, as balancing selection may be caused by sexual dimorphism. Overall, our results suggest that a complex interaction between sex, environmental context (starvation, pathogen exposure – both systemic and oral), and genotype may contribute to the long-term maintenance of immune alleles.

## Materials and methods

### *Drosophila* lines and rearing

Conventionally reared flies were maintained in a 23°C incubator with a 12 hr light:12 hr dark schedule on a cornmeal-molasses-yeast diet 64.3 g/L cornmeal, 79.7 mL/L molasses, 35.9 g/L yeast, 8 g/L agar, 15.4 mL of food acid mix (50 mL phosphoric acid + 418 mL propionic acid + 532 mL deionized water) and 1 g/L Tegosept. We used CRISPR/Cas9 genome editing to modify the *diptericin A* gene. Briefly, DNA coding for guide RNA was inserted into the pUS-BbsI plasmid (*Supplementary file 1*). A single-stranded donor DNA (120 bp) containing the desired edit (to change from serine to arginine) and a silent mutated PAM site was synthesized by IDT (Coralville, IA, USA). The plasmid and ssODN were then injected into Bloomington stock #55821, which expresses Cas9 driven by the *vasa* promoter, by Genetivision, Inc (Houston, TX, USA). Individual flies developed from the injected embryos were collected and crossed with a modified version of Bloomington stock # 7198 (a line with *CyO/Kruppel* balanced on the second chromosome, and *serrate*/*Dichaete* balanced on the third chromosome). Our version, 7198[A4], was provided by Stuart Macdonald and has the DSPR (*King et al., 2012*) A4 line's X chromosome instead of the *w*[*] from the original 7198. The F1s were collected and again individually crossed with 7198[A4] for F2 crosses, yielding individuals with a homozygous second chromosome representing one of the chromosomes carried by the original injected embryo. The Dpt gene was sequenced from the F2 cross progeny to determine whether edits occurred. This yielded several classes of alleles including homozygous serine *dpt* (wildtype, *dpt^S69^*), homozygous arginine *dpt* (*dpt^S69R^*), and *dpt* null (Δ*dpt,* refers to lines with either 1 or 3 base pair deletion) (*Figure 1—figure supplement 1A*). Balancers were removed, and lines were moved into the same genetic background through a series of crosses as shown in *Figure 1—figure supplement 1B*. An *imd* line was used as an

IMD pathway negative control, but note that this line was from a completely different genetic background from the rest of the lines.

## Axenic fly preparation

Microbe-free (axenic) lines were generated by first washing embryos in a 10% bleach solution to dissolve the chorion for 2 min. The embryos were then washed in 70% ethanol for 30 s and water for another 30 s, then transferred to autoclaved molasses food (see above). Some embryos from each treatment were placed onto De Man, Rogosa, and Sharpe (MRS) agar plates and incubated at 30°C for 48 hr to check that they did not contain viable microbes. Axenic lines were continuously checked for the presence of contaminating microbes (every three to four generations) by homogenizing flies and plating the homogenate on MRS agar.

Axenically and gnotobiotically (see below) reared flies were maintained in an incubator that was isolated from conventionally reared flies. The incubator was kept at 23°C with a 12 hr light:12 hr dark schedule. Axenic and gnotobiotic flies were kept on the same molasses diet that had been autoclaved before dispensing into autoclaved vials. Axenic and gnotobiotic flies were only handled inside a sterile hood (Baker SG 400, The Baker Company Inc, Sanford, ME, USA).

## Bacterial strains

The following bacteria were used for systemic infection assays: *P. rettgeri* (*Juneja and Lazzaro, 2009*), *Providencia burhodegraneria* Strain B (*Juneja and Lazzaro, 2009*), *E. faecalis* (*Juneja and Lazzaro, 2009*), *S. marcescens* (*Lazzaro et al., 2004*), *L. fusiformis* Strain Juneja (*Smith and Unckless, 2018*), and *S. succinus* (isolated from wild *Drosophila,* Unckless lab). Bacteria were grown from glycerol stocks on LB plates at 37°C overnight.

The following bacteria were used in gnotobiotic experiments: *L. plantarum, L. brevis, A. tropicalis*. All these strains were isolated from plating conventionally reared flies on MRS agar in the Unckless lab at the University of Kansas. Individual colonies from plated fly homogenate were grown overnight in MRS for DNA isolation. Bacterial species were identified using Sanger sequencing with the 16S rRNA region primers 27F (AGAGTTTGATCCTGGCTCAG) and 1492R (CGGTTACCTTGTTACGACTT). We also utilized the same *P. rettgeri* as described for systemic infection.

## Survival assays

For systemic infections, individual colonies of bacteria were picked and grown in 2 mL LB broth shaking overnight at 37°C. Bacterial suspensions were then diluted or concentrated to $OD_{600}=0.1$ for *P. rettgeri,* $OD_{600}=10$ for *S. succinus,* $OD_{600}=1.5$ for *E. faecalis*, $OD_{600}=3.5$ for *L. fusiformis*, $OD_{600}=1.0$ for *L. lactis*, and $OD_{600}=4.0$ for *S. marcescens. L. plantarum* was grown in 5 mL MRS at 30°C overnight and was concentrated to $OD_{600}=10$ for systemic infections. To induce systemic infection, 5–9 days of age, conventionally reared flies were pricked in the thorax with a needle dipped in a bacterial suspension (*Khalil et al., 2015*). Infections were done in triplicate with at least 20 flies for each replicate per line for a total of 60 flies per genotype per condition. Flies were incubated at 23°C with a 12 hr light:12 hr dark schedule, and survival was tracked daily for 5 days post infection.

## Gnotobiotic longevity

Axenically reared flies were collected within 24 hr of eclosion. Flies were then kept on sterile food for 2 days before sorting for longevity experiments. To begin longevity experiments, flies were separated into groups of 10 individuals of each sex and put onto sterile food seeded with 50 µL of bacterial suspension at an OD600 of 15±1. For each replicate, we used 2 vials of 10 flies each per sex per line for a total of 20 flies for each sex per genotype for a total of 60 flies across all replicates. Flies were allowed to feed in the inoculated vials for 3 days before being transferred to uninoculated sterile food vials. Flies were flipped to new sterile media every 4–5 days for the remainder of the experiment. Surviving flies were counted every 1–3 days until all flies were dead.

## Gnotobiotic bacterial load

To determine whether microbes became established in the gut, we homogenized flies during and after the exposure and plated the homogenate. We measured bacterial load by inoculating flies in the same manner as gnotobiotic longevity. Flies were separated into groups of 5 females and 5 males per

vial. For 2-day experiments, flies were kept on the seeded food for the entire experiment. For 10- and 20-day experiments, flies were allowed to feed on the seeded food for 3 days before being transferred to sterile food. Flies continued to be transferred to new sterile media every 3–4 days until days 10 or 20 post feeding. After the experimental (2-, 10-, or 20-day) time period, flies were surface-sterilized by washing in 70% ethanol followed by molecular grade water. Flies were separated by sex, and three individuals were homogenized together in 300 µL of sterile 1× PBS, and the homogenate was plated on the appropriate media using a Whitley WASP Touch spiral plater (Don Whitley Scientific, UK). When there were not 3 flies still alive, then all remaining flies of a sex were used and squished in 100 µL of 1× PBS per fly collected instead of 300 µL to keep all samples the same concentration per fly. Counts were adjusted accordingly.

## Feeding rate assays

To measure the amount of food and bacteria consumed by males and females of different genotypes, we made food containing blue dye by adding 11.2 g FCF blue dye (Erioglaucine disodium salt) per liter of food. We used newly eclosed flies (1–2 days post eclosion/14 days post oviposition) and separated sexes (in sterile conditions) and kept flies at a density of 10 flies/vial in fresh food vials. We held these flies in incubators for 1 day, so they would recover from the stress of anesthesia during sexing. To introduce bacteria (or control media) into the food, we pipetted 50 µL of suspension or LB at an $OD_{600}$ of 15 (±1) into each vial and allowed the suspension to absorb into the food for 30–40 min by keeping the vials open inside a sterile hood. We next added the experimental flies and allowed them to feed for 1 hr. After 1 hr, we anesthetized the flies on ice, then rinsed in ethanol and sterile water. Flies were homogenized in 300 µL of 1× PBS with a glass bead (maximum speed for 4 min). The homogenate was centrifuged at 14,000 RPM for 4 min, and 200 µL of the supernatant was used to measure absorbance at 630 nm. Absorbance differences were analyzed using the natural log of absorbance as the response variable with genotype, sex, genotype by sex interaction, and block as independent variables. Due to the blocking structure of the experiment, each treatment (no media control, LB control, *P. rettgeri*) was analyzed separately.

## Desiccation

Desiccation survival assays were performed on $dpt^{S69}$ and $dpt^{S69R}$ adult males 4–7 days post eclosion in conventionally and axenically reared flies. Ten males were placed into an empty vial and closed off with a cotton plug. Each genotype had 5 vials for a total of 50 flies per line per rearing condition in each replicate. The flugs were topped with silica gel (Fisher, #S684) and sealed with parafilm to prevent any moisture from entering the vials. Vials were kept at 24°C on 12 hr day/night cycles. Survival was measured by counting flies hourly until the entire population died. This was repeated once more with 50 flies per line per rearing condition in each trial for a total of 100 flies, with the exception of the *Δdpt* line, which in total only had 30 flies per condition in total across trials.

## Starvation

Starvation survival assays were performed on $dpt^{S69}$ and $dpt^{S69R}$ adult males 4–7 days after eclosion for axenically and conventionally reared flies. Ten males were placed into a vial with autoclaved starvation media (1% agar). The 1% agar was used to starve flies of nutrition but not desiccate them. Vials were kept at 24°C on 12 hr day/night cycles. Survival was measured by counting surviving flies at three 8 hr intervals (8 am, 4 pm, and 12 am) until all flies died. This was repeated twice more for a total of 3 trials with at least 40 flies per line per rearing condition in each trial for a total of at least 120 flies.

## 16S sequencing

Flies were reared in the lab in two distinct ways for 16S rRNA sequencing of conventionally reared flies. First, flies were taken from vials of individual genotypes. This allows for any moderate effects of fly genotype to equilibrate over time. Second, flies were taken from cages that were started with heterozygous *diptericin* flies. These cages were started with the F1 progeny from crosses of $dpt^{S69}$ and $dpt^{S69R}$ flies and allowed to continue for three discrete generations before flies were collected for 16S rRNA sequencing. This ensures that the genotypes are exposed to the same microbes, and any differences in microbiome are due to genetic differences manifest in that generation.

We isolated DNA from individual flies using Gentra PureGene Tissue DNA Isolation Kit (QIAGEN #158388, QIAGEN, Germantown, MD, USA) following the manufacturer's instructions. DNA pellets were rehydrated in 10 μL DNA hydration solution. Flies from cage populations were genotyped by using the dpt_cw primer pair (*Supplementary file 1*) followed by a restriction enzyme digest using enzymes that cut DNA sequences for arginine (BccI, NEB # R0704S) or serine (AluI, NEB # R0137S).

DNA concentration was measured with Qubit fluorimeter (Invitrogen). Ten flies from each inbred CRISPR genome-edited line (5 females and 5 males from $dpt^{S69}$, $dpt^{S69R}$, $dpt^{S69R-2}$) (a second homozygous arginine line from CRISPR/Cas9 genome editing), and $\Delta dpt$ (1 base pair deletion) and 10 flies of each genotype from the cage population (5 females and 5 males of SS, SR, RR genotypes) were each brought to a DNA concentration of 5 ng/μL. Libraries were prepped in accordance with the 16S Metagenomic Sequencing Library Preparation protocol from Illumina for the 16S V3/V4 region. Sequencing was performed on the Illumina MiSeq platform using v.3 300 bp paired-end reads. Library preparation and sequencing were performed at the University of Kansas Genome Sequencing Core (Lawrence, KS, USA).

Wild flies were collected from decaying apples and pears in an apple orchard in Kansas City, KS, USA (3341N 139th St, Kansas City, KS 66109, USA). Flies were immediately transported back to the lab and sorted by species on $CO_2$ and frozen at –20°C. DNA was extracted from individual flies using Gentra PureGene Cell & Tissue DNA Isolation Kit (QIAGEN #158388). The samples were tested for species (*D. simulans* vs. *D. melanogaster*), *Wolbachia* status, and Dpt genotype using primers listed in *Supplementary file 1* . Collections are summarized in *Supplementary file 10*. Libraries were prepped in the same manner as the conventionally reared flies.

## 16S bioinformatic analysis

Demultiplexed reads were processed with QIIME2, v.2019.10 (*Bolyen et al., 2019*). Primers were removed from 5' ends with Cutadapt using default parameters (*Martin, 2011*). Reads were de-noised and trimmed for quality with Divisive Amplicon Denoising Algorithm (DADA2) within the QIIME2 bioinformatics pipeline (*Callahan et al., 2016*). Forward and reverse reads were truncated at 280 bp and 245 bp, respectively. The remaining ASV table was exported from QIIME2 for further processing in R (*R Development Core Team, 2021*). Taxonomy was assigned to the ASV table using SILVA 16S rRNA gene reference database, v.138 (*Quast et al., 2013*; *Yilmaz et al., 2014*; *Glöckner et al., 2017*). Reads assigned to genus level *Wolbachia* (a *Drosophila* endosymbiont) and kingdom level Eukaryota were removed from further analysis. We also removed reads not observed at least three times in at least 10 samples. Then, conventionally reared flies were rarefied to 17,066 reads, while wild flies were rarefied to 10,771 reads.

All statistical analysis of 16S data was performed in R (4.1.2) using the Phyloseq (*McMurdie and Holmes, 2013*) package and ggplot2 (*Wickham et al., 2019*) package for visualization. The CRISPR and cage population data were analyzed separately. For each population, alpha diversity was estimated using Shannon diversity in Phyloseq using the estimate_richness() function. Bray-Curtis dissimilarity was calculated to look at overall patterns of microbiome composition in Phyloseq using the ordinate() function. Significance was determined at α=0.05. Fixed effects models were fit with the package lme4 (*Bates et al., 2015*) with the fixed effects genotype + sex + genotype*sex. When necessary, p-values were adjusted for multiple comparisons using the FDR correction method.

## Statistical analysis

R (v.4.1.2) was used to run statistical analyses. Survival data were plotted using the R package survminer (*Kassambara et al., 2021*). The analysis was performed using the Cox proportional hazards regression model in R (*Therneau, 2009*). For longevity, significance was determined using the model: Lifespan~(Genotype*Sex)/Vial + Block. For gnotobiotic bacterial load, significance was determined using the model: Colonies.mL~(Genotype)*Sex + Block.

## Material availability statement

All edited *Drosophila* lines and microbes are available from the authors upon request.

## Acknowledgements

We thank the Cider Hill Family Orchard for allowing us to collect wild flies on their property; S Macdonald lab for providing the modified 7198 and A4 lines; Kistie Brunsell for help with DNA extractions of wild flies; and Elizabeth Everman for assistance with the feeding assay. We also thank two anonymous reviewers for helpful comments. The University of Kansas Genome Sequencing Center Core (supported by NIH CMADP COBRE P20-GM103638) performed 16S sequencing. SRM was supported by a University of Kansas Self Graduate Fellowship, and the work was supported by NIH R01-AI139154 to RLU.

## Additional information

### Funding

| Funder | Grant reference number | Author |
| --- | --- | --- |
| National Institutes of Health | AI139154 | Robert L Unckless |
| National Science Foundation | 2330095 | Robert L Unckless |
| National Institutes of Health | CMADP COBRE P20-GM103638 | Robert L Unckless |
| University of Kansas Self Graduate Fellowship | | Sarah R Mullinax |

The funders had no role in study design, data collection and interpretation, or the decision to submit the work for publication.

### Author contributions

Sarah R Mullinax, Conceptualization, Formal analysis, Validation, Investigation, Methodology, Writing – original draft; Andrea M Darby, Anjali Gupta, Investigation, Methodology; Patrick Chan, Investigation; Brittny R Smith, Resources, Investigation; Robert L Unckless, Conceptualization, Data curation, Formal analysis, Supervision, Funding acquisition, Validation, Investigation, Visualization, Methodology, Writing – original draft, Project administration, Writing – review and editing

### Author ORCIDs

Anjali Gupta ⬤ https://orcid.org/0000-0002-5717-543X
Robert L Unckless ⬤ https://orcid.org/0000-0001-8586-7137

Reviewer #1 (Public review): https://doi.org/10.7554/eLife.90638.3.sa1
Author response https://doi.org/10.7554/eLife.90638.3.sa2

## Additional files

### Supplementary files

MDAR checklist

Supplementary file 1. Oligonucleotide sequences used in this study.

Supplementary file 2. Cox proportional hazard pairwise significance p-values for systemic infection. p-Values less than 0.05 are underlined.

Supplementary file 3. Cox proportional hazard ratio pairwise p-values for systemic infections with males and females. p-Values less than 0.05 are underlined.

Supplementary file 4. p-Values from Tukey's HSD for axenic and gnotobiotic longevity. p-Values less than 0.05 are underlined.

Supplementary file 5. p-Values from Tukey's HSD for bacteria load. p-Values less than 0.05 are underlined.

Supplementary file 6. p-Values from Tukey's HSD for bacterial load from poly-association with *L. plantarum* and *P. rettgeri*. p-Values less than 0.05 are underlined.

Supplementary file 7. Desiccation resistance Tukey's HSD p-values. p-Values less than 0.05 are underlined. 'A' refers to flies reared axenically, 'C' refers to flies reared on a conventional diet.

Supplementary file 8. Starvation resistance Tukey's HSD p-values. p-Values less than 0.05 are underlined. 'A' refers to flies reared axenically, 'C' refers to flies reared on a conventional diet.

Supplementary file 9. Conventional longevity Cox proportional hazards regression p-values. p-Values less than 0.05 are underlined. 'F' and 'M' refer to males and females, respectively; 'het' refers to heterozygotes for serine/arginine.

Supplementary file 10. Genotyped wild-caught *D. melanogaster*. Counts of serine/arginine polymorphism genotypes for wild-caught *D. melanogaster* including *Wolbachia* infection status and sex.

### Data availability

Sequencing data have been deposited under bioproject PRJNA1255302. All other data are at Dryad.

The following datasets were generated:

| Author(s) | Year | Dataset title | Dataset URL | Database and Identifier |
|---|---|---|---|---|
| Mullinax SR, Darby AM, Gupta A, Chan P, Smith BR, Unckless RL | 2025 | A suite of selective pressures supports the maintenance of alleles of a *Drosophila* immune peptide | https://datadryad.org/dataset/doi:10.5061/dryad.qz612jmt1 | Dryad Digital Repository, 10.5061/dryad.qz612jmt1 |
| Mullinax SR, Darby AM, Gupta A, Chan P, Smith BR, Unckless RL | 2025 | A suite of selective pressures supports the maintenance of alleles of a *Drosophila* immune peptide | https://www.ncbi.nlm.nih.gov/bioproject/PRJNA1255302 | NCBI BioProject, PRJNA1255302 |

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
