## [Editor Report · eLife Assessment]

This **valuable** study investigates evolutionary aspects around a single amino acid polymorphism, known to be under long-term balancing selection, in an immune peptide of *Drosophila melanogaster*. Using alleles with different substitutions, the investigators demonstrate that while one allele provides better survival after systemic infections by a bacterial pathogen, the alternative allele endows its carriers with a longer lifespan under certain conditions. The authors suggest that these contrasting fitness effects of the two alleles contribute to balancing their long-term evolutionary fate. While the work is very interesting, the strength of the provided evidence is still **incomplete**, and the study would benefit from more rigorous approaches.

---

## [Referee Report · Reviewer #1 (Public review)]

Summary:

In this manuscript, Unckless and colleagues address the issue of the maintenance of genetic diversity of the gene diptericin A, which encodes an antimicrobial peptide in the model organism *Drosophila melanogaster*. This is an important question as the maintenance of different alleles in wild populations is not known.

Strengths:

The data indicate that flies homozygous for the dptA S69 allele are better protected against some bacteria. By contrast, male flies homozygous for the R69 allele resist better to starvation than flies homozygous for the S69 allele. This provides an element of explanation.

Weaknesses:

(1) Some of the results are difficult to understand. The observation that R69 die more than the null Dpt mutant and the wild-type is strange. This could be due to background effect. The fact that the second chromosome was not isogenized after the CRISPR change is an issue. This issue may take too much time to fix, but should be acknowledged. The existence of background effect and the multiple tested conditions that may lead to the obtention of results that may not be reproduced in other contexts/labs.

(2) Some lifespans are rather short and often in disagreement with other studies (Leulier, Iatsenko but also Hanson/Lemaitre). There are also disagreements inside the article itself for instance between Fig4C and 2A. This should be mentioned.

(3) The shape of many lifespan analysis with abrupt decline contrast with classical lifespan studies, suggesting technical problems.

---

## [Author Response]

The following is the authors’ response to the original reviews.

**Public Reviews:**

**Reviewer #1 (Public Review):**
Summary:In this manuscript, Unckless and colleagues address the issue of the maintenance of genetic diversity of the gene diptericin A, which encodes an antimicrobial peptide in the model organism *Drosophila melanogaster*.Strengths:The data indicate that flies homozygous for the dptA S69 allele are better protected against some bacteria. By contrast, male flies homozygous for the R69 allele better resist starvation than flies homozygous for the S69 allele.Weaknesses:-I am surprised by the inconsistency between the data presented in Fig. 1A and Fig. S2A for the survival of male flies after infection with P. rettgeri. I am not convinced that the data presented support the claim that females have lower survival rates than males when infected with P. rettgeri (lines 176-182).

The two figures are pasted above (1A left, S2A right). The reviewer is correct that the two experiments look different in terms of overall outcomes for males, though qualitatively similar. These two experiments were performed by different researchers, and as much as we attempt to infect consistently from researcher to researcher, some have heavier hands than others. It is true that the genotype that has the largest sex effect is the arginine line (blue) where females (in this experiment) are as bad as the null allele, and males are more intermediate. Also note that the experiments in S2A (male and female) were done in the same block so they are the better comparison. We’ve reflected this in the manuscript.

- The data in Fig. 2 do not seem to support the claim that female flies with either the dptA S69 or the R69 alleles have a longer lifespan than males (lines 211-215). A comment on the [delta] dpt line, which is one of the CRISPR edited lines, would be welcome.

We’ve reworded this section based on these comments.

- The data in Fig. 2B show that male flies with the dptA S69 or R69 alleles have the same lifespan when poly-associated with L. plantarum and A. tropicalis, which contradicts the claim of the authors (lines 256-260).

This is correct – the effect is only in females. It has been corrected.

**Reviewer #2 (Public Review):**
Summary: In this study, the authors delve into the mechanisms responsible for the maintenance of two diptericin alleles within *Drosophila* populations. Diptericin is a significant antimicrobial peptide that plays a dual role in fly defense against systemic bacterial infections and in shaping the gut bacterial community, contributing to gut homeostasis.Strengths: The study unquestionably demonstrates the distinct functions of these two diptericin alleles in responding to systemic infections caused by specific bacteria and in regulating gut homeostasis and fly physiology. Notably, these effects vary between male and female flies.Weaknesses: Although the findings are highly intriguing and shed light on crucial mechanisms contributing to the preservation of both diptericin alleles in fly populations, a more comprehensive investigation is warranted to dissect the selection mechanisms at play, particularly concerning diptericin's roles in systemic infection and gut homeostasis. Unfortunately, the results from the association study conducted on wild-caught flies lack conclusive evidence.

This is true that the wild fly association study is mostly a negative result. We’ve backed off the claim about the *Morganella* association.

Major Concerns:Lines 120-134: The second hypothesis is not adequately defined or articulated. Please revise it to provide more clarity. Additionally, it should be explicitly stated that the first part of the first hypothesis (pathogen specificity), i.e., the superior survival of the S allele in Providencia infections compared to the R allele, has been previously investigated and supported by the results in the Unckless et al. 2016 paper. The current study aims to additionally investigate the opposite scenario: whether the R allele exhibits better survival in a different infection. Please consider revising to emphasize this point.

We’ve reworded this section and added references to both the Unckless et al. 2016 and Hanson et al. 2023 papers.

Figures and statistical analyses: It is essential to present the results of significant differences from the statistical analyses within Figures 1B, 2B, and 3. Additionally, please include detailed descriptions of the statistical analysis methods in the figure legends. Specify whether the error bars represent standard error or standard deviation, particularly in Figure 3, where assays were conducted with as few as 3 flies.

We have added statistical details as requested.

Lines 317-318 (as well as 320-328): The data related to P. rettgeri appear somewhat incomplete, and the authors acknowledge that bacterial load varies significantly, and this bacterium establishes poorly in the gut. These data may introduce more noise than clarity to the study. Please consider revising these sections by either providing more data, refining the presentation, or possibly removing them altogether.

The fact that *P. rettgeri* establishes poorly in the gut in wildtype flies is the result of several unpublished experiments in the Lazzaro and Unckless labs. We don’t have this as a figure because it was not directly tested in these experiments. We’ve added a note that it is personal observation and we’ve reworked the discussion in the second section.

Lines 335-387 and Figure 4: Although these results are intriguing and suggest interactions between functional diptericin and fly physiology, some mediated by the gut microbiome, they remain descriptive and do not significantly contribute to our understanding of the mechanism that maintains the diptericin alleles.

While the reviewer is correct that these experiments do not elucidate mechanism, they do strongly suggest (based on the controlled nature of the experiments) that the physiological tradeoffs are *due* to *Diptericin* genotype. The disagreement is the level of “mechanism”. At the evolutionary level, the demonstration of a physiological cost of a protective immune allele is sufficient to explain the maintenance of alleles. However, we have not determined (and did not attempt to determine) why Diptericin genotype influences these traits. That will have to wait for future experiments.

Lines 399-400: The contrast between this result and statement and the highly reproducible data presented in Figures 2-4 should be discussed.

We’ve added some discussion to this section including a reference to the “inconstancy” of the *Drosophila* gut microbiome.

Lines 422-429 and Figure 5D: The conclusion regarding an association between diptericin alleles and Morganellaceae bacteria is not clearly supported by Figure 5D and lacks statistical evidence.

We’ve changed this to just be suggestive.

**Reviewer #3 (Public Review):**
Summary:This paper investigates the evolutionary aspects around a single amino acid polymorphism in an immune peptide (the antimicrobial peptide Diptericin A) of *Drosophila melanogaster.* This polymorphism was shown in an earlier population genetic study to be under long-term balancing selection. Using flies with different AA at this immune peptide it was found that one allelic form provides better survival of systemic infections by a bacterial pathogen, but that the alternative allele provides its carriers a longer lifespan under certain conditions (depending on the microbiota). It is suggested that these contrasting fitness effects of the two alleles contribute to balance their long-term evolutionary fate.Strengths:The approach taken and the results presented are interesting and show the way forward for studying such polymorphisms experimentally.Weaknesses:(1) A clear demonstration (in one experiment) that the antagonistic effect of the two selection pressures isolated is not provided.The study is overwhelming with many experiments and countless statistical tests. The overall conclusion of the many experiments and tests suggests that "dptS69 flies survive systemic infection better, while dptS69R flies survive some opportunistic gut infections better." (line 444-446). Given the number of results, different experiments, and hundreds of tests conducted, how can we make sure that the result is not just one of many possible combinations? I suggest experimentally testing this conclusion in one experiment (one may call this the "killer-experiment") with the relevant treatments being conducted at the same time, side by side, and the appropriate statistical test being conducted by a statistical test for a treatment x genotype interaction effect.

This is a nice idea but would not work in practice since the fly lines used are different (gnotobiotic vs conventional) and gnotobiotics have to be derived from axenic lines that need a few generations to recover from the bleaching treatment.

(2) The implication that the two forms of selection acting on the immune peptide are maintained by balancing selection is not supported.The picture presented about how balancing selection is working is rather simplistic and not convincing. In particular, it is not distinguished between fluctuating selection (FL) and balancing selection (BL). BL is the result of negative frequency-dependent selection. It may act within populations (e.g. Red Queen type processes, mating types) or between populations (local adaptation). FL is a process that is sometimes suggested to produce BL, but this is only the case when selection is negative frequency dependent. In most cases, FL does not lead to BL.The presented study is introduced with a framework of BL, but the aspects investigated are all better described as FL (as the title says: "A suite of selective pressures ..."). The two models presented in the introduction (lines 62 to 69; two pathogens, cost of resistance) are both examples for FL, not for BL.

We’ve added a discussion of how fluctuating selection and balancing selection relate at the end of the discussion.

Finally, no evidence is presented that the different selection pressures suggested to select on the different allelic forms of the immune peptide are acting to produce a pattern of negative frequency dependence.

We are not arguing for negative frequency dependent selection. We assume throughout that Dpt allele does not drive overall frequency of *P. rettgeri* in populations since it is a ubiquitous microbe. So evolution within *D. melanogaster* therefore has little to no effect on density of the pathogen.

**Recommendations for the authors:**

**Reviewer #2 (Recommendations For The Authors):**
Minor Comments:Line 31: Rewrite the sentence mentioning "homozygous serine" for improved clarity, especially since the S/R polymorphism of Diptericin has not been introduced yet.

This has been changed to be vague in terms of specific alleles and just refers to “one allele” vs the other.

Lines 87-94: Consider reorganizing this paragraph to maintain a logical flow of the discussion on the *Drosophila* immune system and the IMD pathway.

We explored other orders, but we think that as is (IMD to AMPs in general to AMPs in *Drosophila*) makes the most sense here.

Line 99: Provide an explanation of balancing selection for a broader readership, differentiating it from other modes of selection.

We added a brief discussion but note that the intro has significant discussion of balancing selection.

Lines 105-106: Please provide a proper reference. Additionally, ensure that the Unkless et al. 2016 paper is correctly referenced, both in lines 111 and 138-141.

This has been added.

Lines 138-141: It would be beneficial to state that the previous study by Unkless et al. 2016 did not control for genetic background, which is why the assay was redone with gene editing.

This has been added.

Lines 296-303: Clarify the source of the survival observations and consider incorporating this data into Figure 2 for improved visualization.

We’ve clarified that this is Figure 2.

Lines 390-394: Explain the distinctions between vials and cages, particularly in terms of food consumption, exposure to bacteria, etc., which can be relevant to gut homeostasis.

We’ve added a discussion of why these two approaches are complementary.

**Reviewer #3 (Recommendations For The Authors):**
StatisticsStatistical results are limited to the presentation of p-values (several hundred of them!). For a proper assessment of the statistical analyses, one would also want to see the models used and the test statistics obtained.The statistical tests done are often unclear. For example, in several experiments, pools of 3 trials (blocs) of multiple animals were tested. The blocs need to be included in the model. Likewise, it seems that multiple delta-dpt fly genotypes were produced. Apparently, they were not distinguished later. Were they considered in the statistical analyses? By contrast, two lines of dptS69R flies were reported to show differences. What concept was applied to test for line difference in some cases and not in others?In the same dataset (i.e. data resulting from one experiment), it seems that mostly multiple tests were done. For example, in one case each treatment was contrasted to the dptS69 flies. It is generally not acceptable to break down one dataset in multiple subsets and conduct tests with each subtest. One single model for each experiment should be done. This may then be followed by post-hoc tests to see which treatments differ from each other.

We’ve attempted to clarify these statistical approaches throughout.

Minor pointsIn the legend of Figure 3 it says: "A monoassociations where each plot represents a different experiment,". This is unclear to me. First, how many plots are there: 3 or 12? Second, what means "experiment"? Are these treatments, or entirely different experiments? How was this statistically taken into account?

We’ve changed this to “different condition” which is clearer. We performed statistical analysis independently for each condition and we’ve now discussed that.

Fig. 5D. It is suggested in the text ("Most intriguing", line 426) and the figure legend that the abundance of Morganellaceae in wild-caught flies differs among genotypes. This is not visible in the figure and not convincingly shown in the text. No stats are given.

We’ve now added that these differences are not significant.

Line 458-461: This sentence is unclear.

We’ve attempted to clarify.

What is a "a traditional adaptive immune system"?

We’ve reworded to “an adaptive immune system”.

There are several typos in the manuscript. Please correct.

We’ve attempted to fix typos throughout.

Bold statements are often without references.

We’ve attempted to add appropriate references throughout.